# E-Cadherin Signaling in Salivary Gland Development and Autoimmunity

**DOI:** 10.3390/jcm11082241

**Published:** 2022-04-17

**Authors:** Margherita Sisto, Domenico Ribatti, Sabrina Lisi

**Affiliations:** Department of Basic Medical Sciences, Neurosciences and Sensory Organs (SMBNOS), Section of Human Anatomy and Histology, University of Bari “Aldo Moro”, 70124 Bari, Italy; domenico.ribatti@uniba.it (D.R.); sabrina.lisi@uniba.it (S.L.)

**Keywords:** E-cadherins, salivary glands, morphogenesis, EMT, Sjögren’s syndrome, autoimmunity

## Abstract

E-cadherin, the major epithelial cadherin, is located in regions of cell–cell contact known as adherens junctions. E-cadherin contributes to the maintenance of the epithelial integrity through homophylic interaction; the cytoplasmic tail of E-cadherin directly binds catenins, forming a dynamic complex that regulates several intracellular signal transduction pathways, including epithelial-to-mesenchymal transition (EMT). Recent progress uncovered a novel and critical role for this adhesion molecule in salivary gland (SG) development and in SG diseases. We summarize the structure and regulation of the E-cadherin gene and transcript in view of the role of this remarkable protein in SG morphogenesis, focusing, in the second part of the review, on altered E-cadherin expression in EMT-mediated SG autoimmunity.

## 1. Introduction to Cadherins

Cadherins are transmembrane or membrane-associated glycoproteins that mediate Ca^2+^-dependent cell–cell adhesion and have mainly been described for their instrumental role during morphogenesis [1]. Cadherins’ functions extend to multiple aspects of morphogenesis, ranging from polarization of simple epithelia to the formation of tissues and organs architecture, the conference of resistance to detachment and the control of cellular tissue organization and cohesion [1]. Cadherins expression occurs through a dynamic process and is regulated by a great number of developmental factors and cellular signals. From the analysis of the sequence similarity, cadherins were divided into five subfamilies: classical types I and II (E-, P-, N- and VE-cadherin), atypical (T-cadherin), desmosomal (desmogleins, desmocollins), protocadherins and cadherin-related proteins [2]. The family of classical cadherins includes E (epithelial)-, N (neural)-, P (placental)-, VE (vascular-endothelial)-, R (retinal)- and K (kidney)-cadherins; among these, E-cadherin is essential for the formation of adherens junctions (AJs) in epithelial cells. E-cadherin mediates strong, homotypic adhesion between neighboring epithelial cells, thereby, safeguarding epithelial barrier integrity [2,3]. The lack of a functional, tight junction and desmosome formation in the absence of E-cadherin emphasizes its central role in the regulation of epithelial cell–cell contacts [2].

## 2. E-Cadherin Discovery

E-cadherin, a type-I cadherin, is generally considered the prototype of all cadherins because of its early identification and its thorough characterization both in normal and in pathological conditions. In 1977, Takeichi [4] proposed the existence of a physiological Ca^2+^-dependent cell–cell adhesion that could explain the adhesive properties of a lung cell line in addition to the more known modality of Ca^2+^-independent agglutination. Takeichi discovered a surface protein of about 150 kDa involved in the Ca^2+^-dependent cell–cell adhesion, reporting, for the first time, the E-cadherin adhesion potential. At the same time, other research groups investigated in this field, reaching results that, only later, were linked together. François Jacob’s group, in 1980, described a specific cell-surface glycoprotein named uvomorulin, the 84 kDa fragment of which was responsible for the Ca^2+^-dependent compaction of mouse embryonal cells [5]. By the use of antibodies against this 84 kDa fragment, cell–cell interactions were perturbated, and the compaction of embryos before implantation was prevented. Using experiments based on subsequent trypsinizations, the same research group deduced that a short-lived precursor was produced by cells from which a stable form of 120 kDa is derived; this 120 kDa protein, in presence of Ca^2+^, was cleaved, giving rise to the 84 kDa active fragment. Electron microscopy revealed that uvomorulin was localized in the intermediate junctions or AJs of intestinal epithelial cells [6], and, nowadays, it is established that the 84 kDa fragment corresponds to the ectodomain of E-cadherin. Concurrently, Wheelock’s group [7] reported the identification and purification of a protein, expressed by epithelial cell lines and tissues, that was named cell-CAM 120/80. This identification was achieved using antibodies directed against an 80 kDa protein that was released into serum-free medium by MCF-7 human breast cancer cells [7,8]. These antibodies caused disruption of cell–cell junctions in mouse epithelial cells and enabled characterization of the cell-surface form of the antigen as a glycoprotein of 120 kDa from which the 84 kDa fragment was released. Complementary studies, performed by Begemann and colleagues [9], demonstrated the presence of a 124 kDa cell adhesion glycoprotein in chicken liver epithelial cells named L-CAM, which was converted into an 81 kDa protein by trypsinization in the presence of Ca^2+^. Interestingly, these antibodies did not affect aggregation of retinal cells expressing R-cadherin instead of E-cadherin. Once all these pioneering studies were reconciled, in 1984, the name “cadherins” was introduced [10] to identify this class of cell–cell adhesion molecule. The prefix “E” (for epithelial) was adopted for cadherin expressed by epithelial cells, and subsequent experiments performed by Takeichi’s group revealed the existence of other cadherins which have distinct cellular expression patterns, such as N- and P-cadherins [11]. Once E-cadherin was definitely individuated as a cell–cell adhesion protein, the subsequent phases led to the cloning of the E-cadherin cDNA [12], the individuation of the tertiary structure of E-cadherin extracellular domain [13], the study of the E-cadherin/catenin complexes [14] and the demonstration of a key role of E-cadherin-mediated regulation of cellular replication [15,16].

## 3. E-Cadherin Structure

E-cadherin is a single-span transmembrane protein. E-cadherin protein precursor is a polypeptide with a short signal sequence for import into the endoplasmic reticulum, a pro-peptide of about 130 amino acid residues (AA) and a mature polypeptide of about 728 AA (Figure 1). The mature E-cadherin contains a transmembrane domain, a cytoplasmic domain of 150 AA and an ectodomain of 550 AA comprising five tandemly repeated domains. Four of these domains are known as cadherin repeats (EC1 to EC4), whereas EC5 is characterized by four conserved cysteines [17]. E-cadherin forms calcium-dependent, homotypic cell–cell adhesion structures known as AJs that mediate intercellular adhesion [18], cell polarity, cell–cell communication, cell survival, cell differentiation and tissue development [19,20,21,22]. The extracellular domain is responsible for homophilic interactions between cadherin molecules expressed at the surface of neighboring cells [17]. Cadherin cytoplasmic tails bind to proteins p120-catenin and β-catenin (alternatively, its homolog γ-catenin in some cell types), while p120-catenin regulates the stability of cadherin–catenin complexes at the plasma membrane [23], and β-catenin interacts with the actin-binding protein α-catenin, which contains an actin-binding domain and physically links AJ complexes to the actin cytoskeleton [23,24]. The integrity of the cadherin–catenin complex and the association with the cytoskeletal actin represent prerequisites for cell–cell adhesion [23,24].

## 4. Development of Submandibular Gland 

The submandibular gland (SMG) development occurs through branching morphogenesis [25,26,27,28,29]. Through the use of comparative studies, it is now known that the development of human and mouse salivary glands (SGs) occurs through the same developmental pattern [30,31]. In the mouse, the first stage of SG morphogenesis shows only the initial thickening of the oral epithelium characterizing the prebud stage, which occurs at embryonic day (E) 11.5 [28,32]. The SMG placode is visible as a localized thickening of the oral epithelium adjacent to the tongue. The epithelial thickening gives rise to the initial bud structure by E12.5. By this time in development, the salivary proof enlarges and invaginates into the underlying mesenchyme, which begins to condense, resulting in the formation of a primary bud linked to the oral surface by a duct that will become the major secretory duct. The cells deriving from the neural crests arrange themselves to surround the epithelial sketches, giving rise to the submandibular parasympathetic ganglia. The signals that initiate this neural–epithelial interaction have not been fully described yet [28,33,34]. By E13, known as the pseudoglandular stage, the final part of the bud grows in size and undergoes rounds of clefting and new bud formation, resulting in approximately 3–5 epithelial buds. The lumen formation already starts at this stage by removing the epithelial cells from the center of the solid stalks through programmed cell death apoptosis [28,32,35]. Branching morphogenesis then progresses, and the majority of the ducts develop a lumen at the canalicular stage from about E15.5. Around E17.5, the branches and terminal buds are delved to form the ductal and acinar system, and, at this point, the terminal bud stage is completed and exhibits differentiated terminal end buds and a presumptive ductal system [32].

## 5. The Pivotal Role for E-Cadherin in Salivary Gland Morphogenesis

Although the E-cadherin adhesion receptor mediates different, acknowledged functions during epithelial branching morphogenesis, relatively little is known of how E-cadherin, in addition to directly mediating intercellular adhesion, impacts the development of salivary acini and ducts. Recent studies showed that, during embryonic SMG morphogenesis, E-cadherin plays a decisive role in determining the differentiation of epithelial progenitor cells into acinar or ductal cells in a specific stage of embryonic development and in guiding the development of glandular structures until maturation [36]. In vitro SMG organogenesis experiments from isolated SMG cells confirmed that E-cadherin is predominantly involved in the structuring of the branching morphogenesis of SGs [37]. Clarifying the mechanism responsible for E-cadherin-mediated SMG development will have important implications for the general understanding of branching morphogenesis in the context of epithelial tissue development. It is now clear that the E12.5 SMG contains two distinct cellular populations that present a different E-cadherin junctional organization, which conditions the subsequent phases of cellular differentiation [38]. The external cellular layer located in contact with the basement membrane consists of closely packed epithelial cells surrounding the polymorphic cells located in the region of the internal glandular bud. The role of E-cadherin in SMG development was investigated by inducing E-cadherin inhibition by the use of both specific antibodies against E-cadherin and siRNAs-mediated E-cadherin gene silencing. These interesting experiments revealed that the disorganized cells in the initial bud express E-cadherin and β-catenin uniformly and diffusely over their surface [26,39]; in addition, another columnar cell population was recognized in the outer layer of the initial bud, in contact with the basement membrane, characterized by distinct E-cadherin junctions, likely to be linked to the columnar morphology. When the glandular bud grows and branches, these highly organized columnar cells remain in the outermost part. Strangely, during the E-cadherin inhibition experiments, the columnar organization of these outer cells was not lost. Probably, the lack of E-cadherin was compensated by N-cadherin, which is highly expressed in these cells. On the contrary, the cells of the inner region of the bud did not present well-structured E-cadherin junctions and also expressed markers typical of ductal cells, suggesting that they were probably destined to give rise to the ducts. The cells that gave rise to the ducts were identifiable as early as E13.5, arranged along the proximal–distal axis and characterized by a large number of F-actin filaments and by the expression of cytokeratin ductal marker K7 [36]. Only later, these ductal precursors acquired defined E-cadherin junctions, first detected at the apical–lateral borders of ductal cells, and appeared coincident with lumen formation [36]. At this stage of glandular development, ZO-1 expression was also detected at sites apical to E-cadherin junctions, suggesting that ductal cells are linked through tight junctions [36,39]. Therefore, a lower expression of E-cadherin in the interior layer of the glandular bud appears to be necessary to ensure cellular rearrangement; when ductal lumens are formed, the presence of E-cadherin appears to be necessary to ensure stabilization of the ducts in the developing gland (Figure 2). Through inhibition studies performed by the use of siRNA and specific antibodies, the fundamental role of E-cadherin junctions in the ductal precursor fate was demonstrated during the lumenization process; they probably act by the modulation of apoptotic cascade [32,40].

## 6. E-Cadherin Localization in Adult Normal Salivary Glands

In normal SGs, E-cadherin is localized to the cell membrane of acinar and ductal cells, similar to the expression observed for the mammary gland. It is interesting to note that the infoldings of the plasma membrane at the basal site of the duct cell are strongly positive. In excretory ducts, high columnar cells show the strongest reaction for E-cadherin at the basal aspect, and ductal basal cells are weakly positive or negative, supporting those two cell types to obtain different cellular functions. It is possible that the stage of cellular differentiation may be a factor in the expression of E-cadherin, and ductal basal cells are possible progenitor cells of salivary gland tumors [41]. The basement membrane zone lacks staining. Modified myoepithelial cells and plasmacytoid cells seem not to express E-cadherin [42].

## 7. The Epithelial-to-Mesenchymal Transition (EMT) Process

Epithelial–mesenchymal transition (EMT) is a reversible cellular program that is known to be crucial for embryogenesis, wound healing and malignant progression [43,44]. During EMT, the epithelial cells lose their junctions, present drastic changes in cell polarity, restructure their cytoskeleton and cell–extracellular matrix interactions are remodeled. This process leads to the detachment of epithelial cells from each other and the underlying cellular membrane [45]. Therefore, the cells undergo changes in the transcriptional programs that specify cell shape and reprogram gene expression which lead to enhanced motility of individual cells, promoting the mesenchymal fate [46]. In this context, the epithelial cells progressively lose their cobblestone, epithelial appearance, co-express epithelial and mesenchymal biomarkers, adopt a spindle shape and transiently acquire a quasi-mesenchymal cell state [47,48]. Interestingly, EMT may be induced to varying extents, producing a wide spectrum of intermediate states (“partial EMT”), and may be reversible through mesenchymal-to-epithelial transitions (MET) [47]. Based on these characteristics, recently, these dynamic processes were widely defined as “epithelial–mesenchymal plasticity” [49]. EMT is regulated at various levels by inflammatory stimuli, including cytokines such as transforming growth factor-β (TGF-β), fibroblast growth factor family, epidermal growth factor and hepatocyte growth factor [44,48,50,51]. These EMT-inducing signals upregulate specific transcription factors (TFs) called EMT-TFs (e.g., Snail, Twist and ZEB) to repress E-cadherin expression and induce mesenchymal gene expression [52]. In line with this, small, non-coding, single-stranded RNAs (microRNAs or miRNAs) act in concert with TFs to modulate the induction or repression of the EMT signaling process. The main initiation signals of EMT are, therefore, represented by downregulation of E-cadherin, the expression of which is decreased during EMT, and the loss of function of this protein promotes the EMT transition. The transcriptional repression of E-cadherin has long been considered a critical step during EMT [53].

## 8. E-Cadherin and EMT

E-cadherin, as one of the most important molecules in cell–cell adhesion of epithelial cells [54], is considered the main effector of EMT and a unique start signal. Therefore, it is also considered a potent tumor suppressor because aberrant regulation of E-cadherin is often found in a multitude of malignant epithelial cancers [55,56]. E-cadherin is important in conserving the epithelial phenotype and regulating homeostasis of tissues by modulating various signaling pathways [56]. Loss of E-cadherin is constantly shown at sites of EMT during development and cancer [56,57], and this event enhances cancer cell invasiveness in vitro and contributes to the transformation of adenoma to carcinoma in animal models [58]. Therefore, the expression level of E-cadherin often is inversely correlated with tumor grade and stage. In some cases, E-cadherin-negative cell lines showed the most devastatingly high levels of tumorigenicity in nude mice. Furthermore, the loss of E-cadherin can be the result of different mechanisms, such as the inactivating mutations of the human E-cadherin gene discovered in about 50% of infiltrating breast carcinomas [58]. Promoter methylation, a type of epigenetic alteration, is considered to be the predominant mechanism of inactivation of the E-cadherin gene. This mechanism has been recognized in many solid tumors; in fact, patients who present inactivation of the E-cadherin gene and altered expression of its protein are considered at high risk of developing diffuse gastric carcinoma and, thus, by these criteria at least, E-cadherin is considered a tumor suppressor gene [58,59]. E-cadherin CDH1 gene promoter possesses several regulatory sequences that mediate CDH1 transcriptional repression in mesenchymal cells, especially during EMT [60]. In addition, the methylation of CpG sites located in the CDH1 enhancers correlates with low gene expression [61]. DNA methylation is catalyzed by DNA (cytosine-5)-methyltransferases (DNMTs) [62]. Recent studies revealed the aberrant hypermethylation of CDH1 in hepatocellular carcinomas [63,64]. This hypermethylation seems to involve the activation of DNMT1, DNMT3A1 and DNMT3A2, and the hypermethylation of CpG sites is significantly associated with gene and protein E-cadherin suppression [65]. More details were provided by Hermann et al., who demonstrated a central role for Snail in the CpG methylation of the E-cadherin promoter through the recruitment of DNMT1 [66]. In salivary adenoid cystic carcinoma (SACC), one of the most common malignant SG neoplasms, a reduction of E-cadherin reactivity was also recorded in the solid variant, especially in the peripheral cells that are more likely to cause metastases. The phenotypical alterations observed in these cells suggest the involvement of the EMT process in the progression of SACCs [65]. Interestingly, the expression levels of circRNAs, member of the non-coding RNA family, were upregulated in cancer tissues of SACC patients; cell transfection techniques, used to inhibit the expression of circRNA members in SACC cell lines, demonstrated that the proliferative, invasive and migratory abilities of SACC cells were significantly decreased, and the EMT process was inhibited, affecting E-cadherin expression [67]. Recent findings highlighted another interesting phenomenon called “cadherin switch”, in which the normal expression of E-cadherin is substituted by the abnormal expression of N- or P-cadherin [55,56,68]. This downregulation of E-cadherin is linked with the release of β-catenin that induces the WNT signaling pathway. There is evidence that the malfunction of the E-cadherin/catenin complex permits the separation of malignant cells from the primary tumor mass, thus, provoking tumor progression and metastasis [69]. Several studies demonstrated that reduced expression of E-cadherin and catenins is critical in the development and progression of human carcinomas [69,70,71], while, on the contrary, E-cadherin alone acts as a suppressor molecule in cancer invasion and metastasis [21]. However, the use of E-cadherin/β-catenin as prognostic markers in SG tumors, for instance, may have no predictive value; Furuse and colleagues [70] demonstrated that such molecules may be immunoexpressed, for example, in healthy SGs, as well as in malignant SG neoplasia, invasive or not. Interestingly, the role of E-cadherin in EMT is still debated, and some authors argue that the loss of E-cadherin is not causal nor a necessity for EMT, and restoration of E-cadherin expression in E-cadherin-negative malignant cells does not reverse the EMT [72]. Nilsson et al. also demonstrated that E-cadherin loss is consequential rather than causal for c-erbB2-induced EMT in non-malignant mammary epithelial cell lines [73]. Loss of E-cadherin alone was demonstrated to be insufficient to trigger the EMT program in non-malignant breast cell lines [74]. In addition, loss of E-cadherin expression seems to be an oversimplification because, surprisingly, several metastases still contain high levels of E-cadherin, and epithelial cells expressing E-cadherin can become invasive and metastasize, notably in patients with prostate cancer [75], ovarian cancer [76] and glioblastoma [77]. Interestingly, the dual role of E-cadherin is possibly due to the existence of two forms of E-cadherin, which are membrane-tethered E-cadherin and soluble E-cadherin (sE-cadherin) [78]. sE-cadherin was initially discovered by Wheelock et al. [7], and subsequent studies were carried out to investigate the propriety of sE-cadherin as a cancer biomarker [79]. sE-cadherin interferes with AJs and promotes invasion and metastasis as a paracrine/autocrine signaling molecule in the progression of various types of cancer such as gastric cancer. Therefore, it induces the activity of a dysintegrin and metalloprotease (ADAM) and matrix metalloproteinases (MMPs), as well as modulates several signaling pathways [77,78,79,80]. Furthermore, interesting studies demonstrated that sE-cadherin is highly expressed in ovarian cancer patients, where sE-cadherin induces tumor angiogenesis via activation of β-catenin and NF-κB signaling, thus, causing a carcinoma metastatic spread [80]. 

## 9. The Role of E-Cadherin in Salivary Gland Pathogenesis: Lesson from Sjögren’s Syndrome

The main aspects related to the organization of epithelia in SGs are relevant to understanding the pathophysiological alterations observed in primary Sjögren’s syndrome (pSS), where the protective function of epithelia is lost. pSS is, essentially, a chronic inflammatory autoimmune epithelitis characterized by complex pathogenesis, that affects mainly the lachrymal glands and SGs [81,82]. In this scenario, E-cadherin, which is the main actor in maintaining epithelial tissue integrity and giving strength to conserve polarization of the epithelial cell layers [83], seems to play an important role in the molecular mechanisms involved in pSS [84,85,86]. Preliminary studies reported that tight junction proteins and AJs are downregulated in human minor SGs with pSS, thus, determining a marked disorganization of the apical pole of these cells in pSS patients [87]. Nevertheless, in pSS SGs, lymphocytes invade the epithelial tissue, and this invasion causes a dramatic, decreased exocrine secretion that leads to dry mouth [88]. In fact, the interactions between lymphocytes and the salivary epithelium could potentially determine the loss of glandular tissue and might compromise the epithelial integrity [89]. In this context, recent findings highlighted that the arrangement of apical factors in points proximal and distal to lymphocytic infiltration in SGs remained intact in a mouse model of pSS [89]. It was observed that E-cadherin distribution remained intact in areas without lymphocytic infiltration, while E-cadherin immunoexpression was absent in areas presenting infiltrating lymphocytes, so, contributed to the loss of glandular tissue organization [89]. Altered expression of E-cadherin also seems to have had a fundamental role in the recent line of research that studied the phenomenon of the EMT-dependent fibrosis observed in pSS SGs [84,85,86]. An increased expression of proinflammatory cytokines, such as IL-6, IL-17 and IL-22, in pSS has a pivotal role in the development of EMT-dependent SG fibrosis characterized by the progressive loss of E-cadherin and by the growing increased expression of the mesenchymal markers by SG cells accompanied by dramatic morphological changes [84,85,86]. These studies elucidated that, in pSS SGs’ inflammatory microenvironment, increased expression of TGF-β determines the activation of EMT involving both the canonical SMAD2/3 pathway and the non-canonical MAPK pathway [84,85,86] (Figure 3). These discoveries were enriched by a recent study investigating serum levels of sE-cadherin in relation to infiltrating lymphocytes in pSS to characterize the expression of E-cadherin and integrin αEβ7/CD103 in the pSS SG epithelium [90]. Interestingly, serum levels of sE-cadherin were significantly increased in pSS compared to controls. In addition, membrane-bound E-cadherin and clusters of αEβ7/CD103-positive cells were located, in particular, in acinar and ductal cells in epithelium tissue both in pSS and controls. These findings indicate a suggestive role for the αEβ7/CD103 and E-cadherin interaction in pSS SGs, and the sE-cadherin fragment may also play a role in the tissue destruction, resulting, thus, in the accumulation of fibrotic SG tissue and pSS disease progression [90].

## 10. Conclusions

Unquestionably, E-cadherin is deeply involved in establishing cell polarity and differentiation and, thereby, in the establishment and maintenance of tissue homeostasis during the SGs’ development. In this review, we sought to discuss the impact of mechanisms of E-cadherin on SG morphogenesis. Several parameters can contribute to differences in cell adhesion energies, including, but not limited to, the intrinsic, biophysical properties of E-cadherin bonds and E-cadherin surface expression levels. A key question, thus, remains as to whether cell segregation during SG development can be explained solely in terms of the intrinsic properties of the E-cadherin ectodomain or whether it is also necessary to incorporate cellular properties, including biomechanics and functional responses to E-cadherin ligation. Providing a picture of these interactions proposes many interesting future research avenues to consider. Since E-cadherin is the major determinant of the epithelial phenotype, it represents the main driver of the EMT program, and the characterization of E-cadherin multifaceted expression corroborates the interpretation of E-cadherin’s roles during the EMT activation cascade. Only the codifying of its expression in relation to the cell phenotype and the timing of its loss during the transition of normal ductal epithelium versus the de-differentiated mesenchymal-like state will allow us to better understand the molecular mechanisms in terms of chronic inflammatory diseases such as autoimmune diseases.

## Figures and Tables

**Figure 1 jcm-11-02241-f001:**
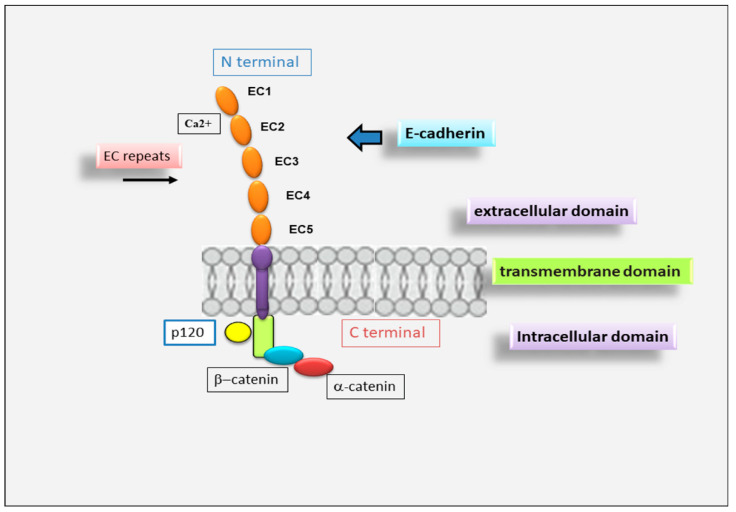
Schematic representation of E-cadherin protein. E-cadherin contains 5 extracellular cadherin (EC) repeats linked by Ca^2+^ binding sites, a transmembrane domain and an intracellular domain that binds p120-α-catenin and β-catenin.

**Figure 2 jcm-11-02241-f002:**
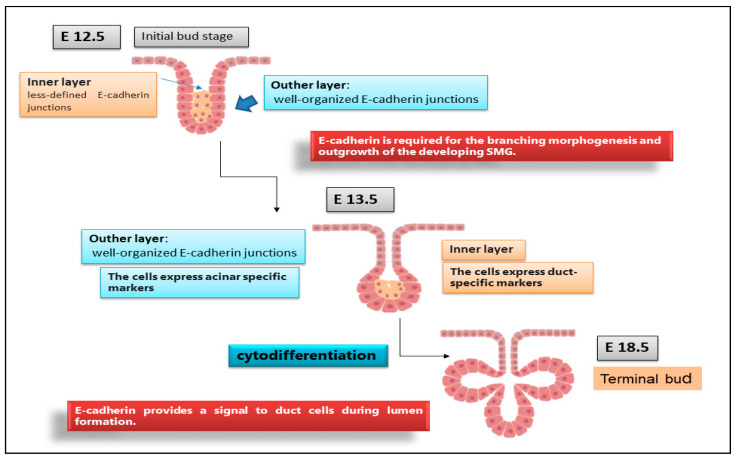
Organization of E-cadherin junctions during SGs morphogenesis. E12.5, E13.5, E18.5 represent stages of SGs embryonic development. At the initial E12.5, E-cadherin was localized to the lateral surfaces of the columnar cells that comprised the outer layer while, in the interior cells, was diffuse, indicating that these cells have less organized E-cadherin junctional structures. By E13.5, outer cell layer expressed a biochemical acinar marker demonstrating that acinar cells begin to differentiate very early in SGs development. The acinar progenitor layer completes the cytodifferentiation at E18.5, expressing E-cadherin in the peripheral cell layer.

**Figure 3 jcm-11-02241-f003:**
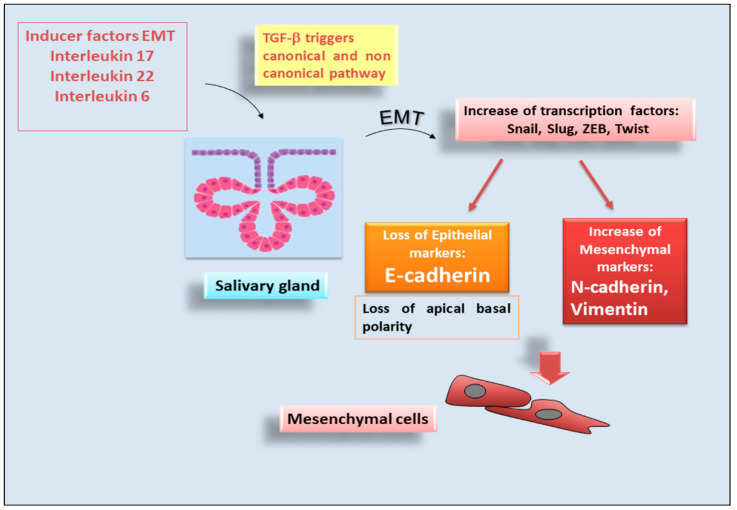
Role of E-cadherin during the epithelial-mesenchymal transition (EMT) process in pSS. In pSS, the transition of epithelial cells versus a mesenchymal phenotype is triggered by several proinflammatory factors and is characterized by the loss of cellular contact and cellular polarity. During EMT, the loss of epithelial marker E-cadherin and an increase of mesenchymal markers occur, through the upregulation of transcriptional factors [i.e., SNAIL, TWIST, Zinc finger E-box-binding homeobox (ZEB), Slug]. The acquisition of mesenchymal markers led to the stabilization of the newly acquired phenotype.

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
