# Peer review of "E-Cadherin Signaling in Salivary Gland Development and Autoimmunity"

_jcm, 2022, doi:10.3390/jcm11082241_

Round 1
Reviewer 1 Report
In this manuscript, the authors reviewed the structure and regulation of the E-cadherin gene and transcript in view of the role of this protein in SGs morphogenesis and development. Overall, the topic is interesting and the article presents most of the information well, but there are still some points should be addressed before it can be accepted for publication.
- The regulation of E-cadherin by DNA methylation in tumorigenesis and EMT should be discusses more detail with more evidence.
- The role of E-Cadherin in the development, metastasis of SACC can be discussed further.
- Consider adding new studies and references from 2021 and 2022.
- In Figure 1, it should be β-Catenin instead of B-Catenin.
Author Response
Manuscript ID: ijms-1655373
Title: E-cadherins signaling in salivary glands development and autoimmunity
Authors: Margherita Sisto, Domenico Ribatti, Sabrina Lisi
We would like to express our sincere gratitude to the reviewer for his constructive and positive comments and for the very thoughtful critique of our manuscript and are pleased to say that we tried to address all the concerns raised. All changes to the manuscript are highlighted in the text. We respond below in detail to each of the reviewer’s comments and we hope that the reviewer will find satisfactory our responses to his comments.
REVIEWER 1
- The regulation of E-cadherin by DNA methylation in tumorigenesis and EMT should be discusses more detail with more evidence.
- The role of E-Cadherin in the development, metastasis of SACC can be discussed further.
We have discussed in more detail both the effect of DNA methylation on E-cadherin expression and the role of E-cadherin in SACC. However, we have not thought of going too far in order not to create a disproportion with the other topics covered in the paragraph. This is a highly developed field of research in the literature and the paragraph aims to offer a broad range of topics for thought on the various mechanisms involved.
- Consider adding new studies and references from 2021 and 2022.
As required, we add studies published in 2021 and 2022.
- In Figure 1, it should be β-Catenin instead of B-Catenin
Done
Reviewer 2 Report
This review manuscript is well written and organized with the information in detail for E-cadherins signaling in the salivary glands.
Please correct a few typos.
Line 65: Please add references.
Line 282: Please remove comma.
Line 274 and after: Please do not use a term of primary and secondary Sjogren's syndrome and use each specific disease name.
Author Response
Thank you very much for your suggestions. I have corrected the manuscript according to your comments.

Reviewer 3 Report
The authors review the role of E-cadherin signaling in salivary gland development and autoimmune disease, especially Sjögren's syndrome.
The reviewer noticed the following minor considerations.
[minor]
Regarding the Greek alphabet: The reviewer finds there are unchanged alphabet characters, which should be in the Greek alphabet.
Figure 1: "B catenin" would be "β-catenin"
Line 306: "integrin alphaEbeta/CD103" would be "integrin αEβ/CD103"
Figure 3: "TGF B" would be "TGF β"
Line 56: There are parentheses without numbers. Please add the appropriate number of referenced papers.
Line 184–6: It is well-known that both "high-columnar cells" and "ductal basal cells" have their specific biological roles. Thus, the reviewer recommends using "supporting" instead of "suggesting."
Author Response
REVIEWER 2
Manuscript ID: ijms-1655373
Title: E-cadherins signaling in salivary glands development and autoimmunity
Authors: Margherita Sisto, Domenico Ribatti, Sabrina Lisi
We would like to express our sincere gratitude to the reviewer for his constructive and positive revision of the manuscript. All minor changes were done into the manuscript.
Regarding the Greek alphabet: The reviewer finds there are unchanged alphabet characters, which should be in the Greek alphabet.
Figure 1: "B catenin" would be "β-catenin"
Line 306: "integrin alphaEbeta/CD103" would be "integrin αEβ/CD103"
Figure 3: "TGF B" would be "TGF β"
Line 56: There are parentheses without numbers. Please add the appropriate number of referenced papers.
Line 184–6: It is well-known that both "high-columnar cells" and "ductal basal cells" have their specific biological roles. Thus, the reviewer recommends using "supporting" instead of "suggesting."
All these minor changes were made into the manuscript and in figures 1, 3.